# Synergistic Antimicrobial Effects of Ibuprofen Combined with Standard-of-Care Antibiotics against Cystic Fibrosis Pathogens

Qingquan Chen, Marleini Ilanga, Sabona B. Simbassa, Bhagath Chirra, Kush N. Shah and Carolyn L. Cannon * 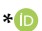

Department of Microbial Pathogenesis and Immunology, Texas A&M University Health Science Center, Bryan, TX 77807, USA; kevinqc@stanford.edu (Q.C.)
* Correspondence: carolyn.cannon@tamu.edu

**Abstract:** Cystic fibrosis (CF) is a common life-shortening genetic disease caused by mutations in the cystic fibrosis transmembrane conductance regulator (CFTR) gene. Lungs of CF patients are often colonized or infected with microorganisms requiring frequent courses of antibiotics. Antibiotic-resistant bacterial infections have been a growing concern in CF patients. Chronic bacterial infections and concomitant airway inflammation damage the lungs, ultimately leading to respiratory failure. Several clinical trials have demonstrated that high-dose ibuprofen reduces the rate of pulmonary function decline in CF patients. This beneficial effect has been attributed to the anti-inflammatory properties of ibuprofen. Previously, we have confirmed that high-dose ibuprofen demonstrates antimicrobial activity against *P. aeruginosa* both in vitro and in vivo. However, no study has examined the antimicrobial effect of combining ibuprofen with standard-of-care antimicrobials. Here, we evaluated the possible synergistic activity of combinations of common nonsteroidal anti-inflammatory drugs (NSAIDs), namely, ibuprofen, naproxen, and aspirin, with commonly used antibiotics for CF patients. The drug combinations were screened against different CF clinical isolates. Antibiotics that demonstrated increased efficacy in the presence of ibuprofen were further tested for potential synergistic effects between these NSAIDS and antimicrobials. Finally, a survival analysis of a *P. aeruginosa* murine infection model was used to demonstrate the efficacy of the most potent combination identified in in vitro screening. Our results suggest that combinations of ibuprofen with commonly used antibiotics demonstrate synergistic antimicrobial activity against drug-resistant, clinical bacterial strains in vitro. The efficacy of the combination of ceftazidime and ibuprofen against resistant *P. aeruginosa* was demonstrated in an in vivo pneumonia model.

**Keywords:** *Pseudomonas aeruginosa*; NSAIDS; ibuprofen; naproxen; aspirin; synergy; cystic fibrosis; ceftazidime



## 1. Introduction

Chronic infection and inflammation are the hallmarks of cystic fibrosis (CF) lung disease and are responsible for most of the morbidity and mortality in CF patients [1–3]. Chronic infection elicits an acute inflammatory response, which is characterized by a persistent neutrophil influx [4]. However, due to the distinctly altered lung environment in CF patients, the inflammation fails to clear the infection, resulting in excessive airway inflammation and worsened airway obstruction [4,5]. Although airway clearance and antibiotics are the mainstays of CF therapy due to the deleterious effects of inflammation, numerous studies have investigated therapies targeting the excessive inflammatory response [4,6,7]. Several in vivo models and clinical trials have demonstrated the beneficial clinical effects of oral and inhaled corticosteroids, macrolides, and nonsteroidal anti-inflammatory drugs (NSAIDs), such as ibuprofen [8–13].

Because it is effective and safe, ibuprofen is advantageous as an anti-inflammatory drug. Compared to corticosteroids that cause growth retardation and cataracts, ibuprofen

has fewer safety concerns, primarily with mild gastrointestinal (GI) symptoms, although ibuprofen can rarely cause GI bleeding and kidney injury. Ibuprofen has been demonstrated to reduce the recruitment of neutrophils into the airway in both a mouse model of acute *Pseudomonas aeruginosa* pulmonary infection and a rat model of endotoxin-induced alveolitis [14,15]. In a chronic *P. aeruginosa* endobronchial infection study, rats treated orally with ibuprofen resulting in a drug plasma concentration of $55 \pm 24$ μg/mL achieved a significant reduction in the inflammatory response and improved weight gain compared with controls [16]. At the same concentration, ibuprofen reduced the production of leukotriene B4, is a proinflammatory mediator, without reducing the pulmonary bacterial burden [16]. Further, Konstan et al. conducted a randomized, double-blind, placebo-controlled clinical trial in CF patients to evaluate the safety and efficacy of high-dose ibuprofen (20 to 30 mg ibuprofen per kilogram of body weight, to a maximum of 1600 mg to achieve 50 to 100 μg/mL peak serum concentrations of ibuprofen). High-dose ibuprofen reduced the rate of decline of pulmonary function in CF patients [10]. Patients between 6 and 17 years old with a baseline forced expiratory volume in 1 s (FEV1) of greater than 60% demonstrated a significantly lower rate of decline in the predicted FEV1% when treated with high-dose ibuprofen than patients treated with placebo [17]. Lands et al. also investigated the safety and efficacy of high-dose ibuprofen in pediatric and adolescent CF patients between ages 6 and 18 years [8]. This randomized, multicenter, double-blinded, placebo-controlled trial did not show a statistically significant difference in the mean annual rate of decline in FEV1. However, the annual rate of decline in the forced vital capacity (FVC) percentage predicted significantly decreased in patients treated with high-dose ibuprofen compared to the placebo-treated group [8]. Thus, these clinical trials suggested the benefits and relative safety of long-term use of high-dose ibuprofen in CF patients and attributed these benefits to the anti-inflammatory properties of ibuprofen.

In addition to the anti-inflammatory activity, studies have also documented the antimicrobial and antifungal activity of ibuprofen. Particularly, our lab has shown a direct, dose-dependent antimicrobial effect of high-dose ibuprofen on two of the most important Gram-negative bacterial pathogens (i.e., *P. aeruginosa* and *Burkholderia* spp.) responsible for chronic pulmonary infection in CF patients [18]. In an acute *P. aeruginosa* pneumonia model, mice orally administrated with high-dose ibuprofen showed a reduced bacterial burden in the lung, and superior survival compared to mice with sham treatment [18]. Additionally, several NSAIDs, including ibuprofen and aspirin, are synergistic with cefuroxime and chloramphenicol against methicillin-resistant *Staphylococcus aureus* (MRSA) [19]. We hypothesized that the remarkable results of the CF ibuprofen trials illustrate that, in addition to its anti-inflammatory properties, ibuprofen mitigates lung function decline through the inhibition of bacterial growth. Hence, we propose that ibuprofen and other NSAIDs, such as naproxen and aspirin, will sensitize drug-resistant bacteria to established antimicrobials, thus exerting a synergistic bactericidal effect. The dual anti-inflammatory and antimicrobial activity of ibuprofen, and possibly other NSAIDs, is ideal for treating both the lung hyperinflammation and infection faced by CF patients without incurring the risks seen with other anti-inflammatories. Combining already approved therapeutics for human use will allow rapid development of novel, effective intervention strategies for the treatment of chronic lung infections caused by multi-drug resistant (MDR) pathogens, such as *P. aeruginosa* and *A. xylosoxidans*, found in the lungs of CF patients.

## 2. Material and Methods

### 2.1. Drugs

Ibuprofen, aspirin, and aztreonam are purchased from Tokyo Chemical Industry (TCI Ltd.) Tokyo, Japan. Naproxen is purchased from Spectrum Chemical Mfg. Corp. (New Brunswick, NJ, USA). Ceftazidime is purchased from CHEM-IMPEX INT'L Inc. (Wood Dale, NJ, USA). Amikacin is purchased from Fisher Scientific (Fair Lawn, NJ, USA).

### 2.2. Bacterial Strains

CF isolates of *Pseudomonas aeruginosa*, *Elizabethkingia meningoseptica*, *Achromobacter xylosoxidans*, *Haemophilus influenzae*, and methicillin-resistant *Staphylococcus aureus* (MRSA) were utilized for our studies. The *P. aeruginosa* isolate PA HP3, *E. meningoseptica* isolate EM 2-18, *A. xylosoxidans* isolate AX 2-79, *H. influenzae* isolate 2501, and MRSA isolate LL06 were recovered from the sputa of cystic fibrosis patients at St. Louis Children's Hospital as standard clinical practice and deidentified for studies. *H. influenzae* isolate HI 2501 was kindly provided by Joseph St. Geme (University of Pennsylvania, Philadelphia, PA, USA).

### 2.3. Bacterial Culture

Bacteria were streaked from frozen glycerol stock onto tryptic soy agar (TSA, BD BBL) or chocolate agar (Hardy Diagnostics, Santa Maria, CA, USA) plates and incubated overnight at 37 °C with 5% carbon dioxide ($CO_2$) until individual colonies formed. A single colony was inoculated into 5 mL Mueller Hinton (MH, BD Difco, Franklin Lakes, NJ, USA) or Brain Heart Infusion (BHI, BD Difco, Franklin Lakes, NJ, USA) media and grown at 37 °C in a shaking incubator at 200 rpm to an $OD_{650}$ of 0.4, which corresponds to ~$5 \times 10^8$ colony forming units (CFU)/mL. Bacteria cultures were adjusted to $1 \times 10^6$ CFU/mL to prepare a working stock for all experiments.

### 2.4. Disc Diffusion Assay

MH agar plates were prepared by autoclaving MH with 17 g agar per liter of media. After autoclaving, the agar was cooled to 70 °C, and 100 μg/mL solution of ibuprofen dissolved in DMSO was added to a final concentration of 100 μg/mL. An equivalent amount of DMSO was added to another batch of MH agar to serve as a control. Plates were cast from MH agar, ibuprofen supplemented MH agar, and DMSO supplemented MH agar. A total of 100 μL of the working stock bacterial culture was dispensed onto agar plates and spread evenly, after which antibiotic (amikacin 30 μg, aztreonam 30 μg, ceftazidime 30 μg, colistin 10 μg, tobramycin 10 μg, gentamicin 10 μg, levofloxacin 5 μg, and vancomycin 5 μg)-infused discs were placed on top of the agar. Plates were incubated between 18 and 24 h. Susceptibility was determined by measuring the diameter of the zone of growth inhibition. Experiments were repeated in triplicate and reported in median diameter. The antibiotic content of the discs was in accordance with the recommendations of the Clinical Laboratory Standards Institute (CLSI), except for colistin and vancomycin for which there are no CLSI-recommended standards [20].

### 2.5. Minimum Inhibitory Concentration Assay

Minimum inhibitory concentrations (MIC) were determined according to the standard Clinical and Laboratory Standards Institute (CLSI) broth microdilution method and adapted from previous studies [21]. Briefly, a 100 μL aliquot of a bacterial suspension was added to each well (n = 3) containing 100 μL ibuprofen, naproxen, aspirin, ceftazidime, amikacin, or aztreonam solution in a 96-well plate to achieve $5 \times 10^5$ CFU/well. All solutions were comprised of 95% MH broth and 5% (*v/v*) DMSO. Bacteria were incubated with 0, 0.06, 0.13, 0.25, 0.5, 1, 2, 4, 8, 16, 32, 64, and 128 μg/mL amikacin, aztreonam, or ceftazidime, or 0, 1, 2, 4, 8, 16, 32, 64, 128, 256, 512, and 1024 μg/mL ibuprofen, naproxen, or aspirin at 37 °C for 18–24 h under static conditions. The final concentration of DMSO in the assay was 2.5% (*v/v*). The MIC was determined as the lowest concentration that showed no bacterial growth upon visual inspection. All experiments were performed in triplicate.

### 2.6. Checkerboard Assay

One *P. aeruginosa* isolate (PA HP3) and one *E. meningoseptica* isolate (EM 2-18) were selected for further study based on their susceptibility in the disc diffusion assay. The final drug concentrations of ibuprofen, naproxen, and aspirin were 0, 50, 75, and 100 μg/mL. Based on the MIC values, a dynamic concentration scale for amikacin, aztreonam, and ceftazidime was used to determine the optimal ratio of synergistic concentrations between

the two therapeutic agents. The final drug concentrations of amikacin against EM 2-18 were 0, 1, 2, 4, 8, 12, and 16 μg/mL. The final drug concentrations of aztreonam against PA HP3 were 0, 0.25, 0.5, 1, 2, and 4 μg/mL. The final drug concentrations of ceftazidime against PA HP3 were 0, 1, 2, 4, 8, 12, and 16 μg/mL. The final solutions were comprised of 95% MH broth and 5% DMSO. A 100 μL working stock of bacterial suspension was incubated with a 100 μL solution of therapeutic agents (n = 3) in 96 wells plate for 18–24 h at 37 °C. Wells demonstrating bacterial growth inhibition were identified visually to determine a synergistic MIC. All experiments were performed in duplicate. To evaluate for potential synergy, the fractional inhibitory concentration (FIC) was calculated using the formula below. The FIC is the sum of the ration of the MIC of each agent alone to the MIC of that agent in combination with the other agent. Interpretation is shown in Table 1 [22–24].

$$\text{FIC} = \frac{\text{MIC A combined}}{\text{MIC A}} + \frac{\text{MIC B combined}}{\text{MIC B}}$$

**Table 1.** Interpretation of FIC values to define synergy based on the checkerboard assay.

| Definition | FIC |
|:---:|:---:|
| Synergistic | FIC $\leq$ 0.5 |
| Additive | 0.5 < FIC $\leq$ 1 |
| Indifferent | 1 < FIC $\leq$ 4 |
| Antagonistic | 4 < FIC |

### 2.7. 24 h Endpoint CFU Assay

The potential synergy between combinations of amikacin, aztreonam, or ceftazidime with ibuprofen or naproxen against *P. aeruginosa* and *E. meningoseptica* isolates PA HP3 and EM 2-18 at a final concentration of $10^6$ CFU/mL were determined using a 24 h endpoint CFU study performed in triplicate. The concentrations of ibuprofen and naproxen tested against PA HP3 and EM 2-18 were 0, 50, and 100 μg/mL. The concentrations of aztreonam and ceftazidime in combination with ibuprofen against PA HP3 were 0, 1, 2, 4, and 8, or 0, 2, 4, 8, and 16 μg/mL, respectively. The concentrations of amikacin in combination with ibuprofen tested against EM 2-18 were 0, 2, 4, 8, 12, 16, and 20 μg/mL. The concentrations of aztreonam and ceftazidime in combination with naproxen against PA HP3 were 0, 1, 2, 4, 8, and 12 or 0, 2, 4, 8, 12, 16, and 20 μg/mL, respectively. The tested concentrations of amikacin in combined naproxen against EM 2-18 were 0, 2, 4, 8, 12, 16, and 20 μg/mL. Synergy was defined as a $\geq 2-\log_{10}$ CFU/mL reduction between combined agents and the most effective individual agent at 24 h [25,26]. A 100 μL working stock of bacterial suspension was incubated with 100 μL drug solution (n = 3) in each well of a 96-well plate at 37 °C for 24 h with constant shaking at 100 RPM. The final solutions were comprised of 97.5% MH broth and 2.5% (*v/v*) DMSO. Finally, a 10-fold serial dilution was performed in MH broth with the bacterial suspension from each well, and 50 μL of each dilution was plated onto a blood agar plate. Plates were incubated for 18 h, and colonies were counted to determine the CFU for each condition. The potential synergistic effects were determined as described above. All experiments were performed in duplicate.

### 2.8. Acute Murine P. aeruginosa Lung Infection Model

Male C57BL/6J mice (The Jackson Laboratory, Bar Harbor, ME, USA) aged 5 weeks were used for the acute lung infection studies [27,28], which were approved by the Texas A&M University Institutional Animal Care and Use Committee (IACUC). Mice were weighed and randomly assigned into four groups of 6 mice each and were housed in a barrier facility under pathogen-free conditions until bacterial inoculation. *P. aeruginosa* PA HP3 was grown in LB (LB), pelleted, washed with phosphate-buffered saline (PBS), and resuspended to an $OD_{650}$ of 2.4 in LB corresponding to ~1.3 × $10^{10}$ CFU/mL, as deter-

mined by serial dilution and plating. Following anesthesia via intraperitoneal injection of a ketamine (60 mg/kg) and xylazine (8 mg/kg) cocktail, mice were intranasally inoculated with 75 mL of bacteria inoculum in LB broth at an $LD_{100}$ of ~$1 \times 10^9$ CFU per mouse. After inoculation, mice were individually housed to allow for continuous monitoring of each mouse by infrared cameras. To test the efficacy of combinational therapy against PA HP3, at 2 h post-infection, mice were treated every 8 h subsequently for a maximum of 7 doses. Ibuprofen-treated mice were intraperitoneally injected with 50 µL saline in water and orally administrated with 50 µL of 50:50 strawberry syrup:water ibuprofen suspension mix (0.5 mg ibuprofen). Ceftazidime-treated mice were intraperitoneally injected with 50 µL of 10 mg/mL ceftazidime and orally administered with 50 µL of 50:50 strawberry syrup:water mix. Combination-treated mice were intraperitoneally injected with 50 µL 10 mg/mL ceftazidime and orally administered with 50 µL of 50:50 strawberry syrup:water ibuprofen suspension mix (0.5 mg ibuprofen). Control mice were intraperitoneally injected with 50 µL saline in water and orally administered with 50 µL of 50:50 strawberry syrup:water mix. Infected mice were weighed and assigned a clinical infection score [29,30] every 24 h post-infection. The clinical score is a semi-quantitative metric that evaluates for signs of infection and ranges from asymptomatic (score 0) to moribund (score 6) based on the resting posture (0–2), fur (0–1), and activity level (0–3) of the infected mice. Moribund mice were euthanized with an overdose of ketamine:xylazine followed by cardiac puncture for exsanguination, a method approved by our IACUC (TAMU) and consistent with the recommendations of the Panel on Euthanasia of the American Veterinary Medical Association. The time of euthanasia for mice scored as moribund on the daily inspection was noted as the time of death. Otherwise, the time of death was determined through a careful review of the infrared camera recording of each mouse by an observer blinded to the groups. Mice were monitored for up to 72 h.

*2.9. Statistical Analysis*

All statistics were calculated using JMP pro 13 for Macintosh (SAS Institute, Cary, NC, USA), www.jmp.com (accessed on 15 June 2022). Differences between the treatments were investigated by one-way ANOVA followed by Tukey's multiple comparison test (95% confidence intervals). * Indicates $p \leq 0.05$, ** indicates $p \leq 0.01$, *** indicates $p \leq 0.001$, and **** indicates $p \leq 0.0001$. The in vivo survival curves in the infection model were compared using a Log-rank Mantel–Cox test, whereas the differences in weight change and clinical scores of mice receiving different treatments were analyzed using Kruskal–Wallis one-way ANOVA and Dunn's multiple comparison test. Data were deemed to be significantly different for $p \leq 0.05$.

## 3. Results
### 3.1. Ibuprofen Increased the Zone of Inhibition of CF clinical Isolates in a Disc-Diffusion Assay

We characterized the zone of inhibition around antibiotic-infused discs against CF clinical isolates of several bacterial species: *P. aeruginosa* (PA HP3), *E. meningoseptica* (EM 2-18), *A. xylosoxidans* (AX 2-79), *H. influenzae* (HI 2501), and MRSA (SA LL06) (Table 2). The antimicrobial susceptibilities of these CF clinical isolates were determined according to the Clinical Laboratory Standard Institute (CLSI) breakpoints [31]. All isolates demonstrated resistance to at least two tested antibiotics. Particularly, all strains of Gram-negative species, PA HP3, EM 2-18, AX 2-79, and HI 2501 indicated in orange were non-susceptible to three classes of antibiotics, and thus, met the definition of multi-drug resistant [32]. The *S. aureus* strain (SA LL06) was determined to be methicillin-resistant by the St. Louis Children's Hospital clinical microbiology laboratory. Given the resistance to levofloxacin and vancomycin indicated by the disc-diffusion results, SA LL06 also meets the definition of multi-drug resistant. Next, we tested how high-dose ibuprofen affects the zone of inhibition diameters of different antibiotics. After supplementing the agar with high-dose ibuprofen (100 µg/mL) solubilized in DMSO, the zones of inhibition around aztreonam increased significantly against both PA HP3 and HI 2501 compared with a control plate supplemented with

DMSO (Table 3). Due to the presence of DMSO in the agar, we observed differences in the zone of inhibition diameters (Table 3) for all tested antibiotics against all strains except HI 2051 compared with the previous results obtained with agar lacking DMSO (Table 2). Ceftazidime demonstrated a significant increase in the zone of inhibition against PA HP3 after adding 100 μg/mL ibuprofen. Amikacin demonstrated a significant increase in the zone of inhibition against EA 2-18 and AX 2-79 with the addition of 100 μg/mL ibuprofen (Table 3). Gentamicin and vancomycin, with the addition of 100 μg/mL ibuprofen, showed increases in the median zones of inhibition against SA LL06 (Table 3).

**Table 2.** The zone of inhibition around antibiotic-infused disc against cystic fibrosis pathogens. Sensitivity was established using CLSI standards. Green indicates that the bacteria is sensitive to the tested antimicrobial. Yellow indicates that the sensitivity of a bacteria to the tested antimicrobial is intermediate. Red indicates bacteria is resistant to the tested antimicrobial: amikacin (AMK), aztreonam (ATM), ceftazidime (CAZ), colistin (CST), tobramycin (TOB), gentamicin (GEN), levofloxacin (LVX), and vancomycin (VAN). Orange indicates multi-drug resistant isolates that meet either the CDC definition or that of Magiorakos et al. [32] of non-susceptibility to at least three different classes of antimicrobials [33]. SA LL06 is resistant to methicillin.—indicates not tested.

| Strain | Median Zone of Inhibition Diameter (Unit: mm) | | | | | | | |
| --- | --- | --- | --- | --- | --- | --- | --- | --- |
| | **AMK** | **ATM** | **CAZ** | **CST** | **TOB** | **GEN** | **LVX** | **VAN** |
| PA HP3 | 19 | 18 | 13 | 16 | 6 | - | - | - |
| EM 2-18 | 11 | 6 | 6 | 6 | 6 | - | - | - |
| AX 2-79 | 8 | 7 | 23 | 9 | 6 | - | - | - |
| HI 2501 | 14 | 24 | 24 | 14 | 15 | - | - | - |
| SA LL06 | - | - | - | - | - | 22 | 9 | 14 |

**Table 3.** Supplementing the agar with 100 μg/mL ibuprofen increased the median zone of inhibition around antibiotic-infused discs against tested CF clinical isolates. The concentration of DMSO used to solubilize the ibuprofen was used as a control. * indicates a ≥ 5 mm increase in the zone of inhibition after supplementing with 100 μg/mL ibuprofen.

| | Median Zone of Inhibition Diameter (mm) | | | | | | | | | | | | | | | |
| --- | --- | --- | --- | --- | --- | --- | --- | --- | --- | --- | --- | --- | --- | --- | --- | --- |
| | **AMK** | | **ATM** | | **CAZ** | | **CST** | | **TOB** | | **GEN** | | **LVX** | | **VAN** | |
| DMSO +/− IBU | − | + | − | + | − | + | − | + | − | + | − | + | − | + | − | + |
| PA HP3 | 19 | 21 | 24 | 28 | 13 | 18 * | 14 | 16 | 6 | 6 | - | - | - | - | - | - |
| EM 2-18 | 10 | 15 * | 6 | 6 | 6 | 6 | 6 | 6 | 6 | 6 | - | - | - | - | - | - |
| AX 2-79 | 7 | 9 | 6 | 7 | 22 | 22 | 9 | 9 | 6 | 6 | - | - | - | - | - | - |
| HI 2501 | 14 | 14 | 24 | 29 * | 24 | 26 | 14 | 16 | 15 | 15 | - | - | - | - | - | - |
| SA LL06 | - | - | - | - | - | - | - | - | - | - | 21 | 23 | 7 | 9 | 14 | 15 |

### 3.2. In Vitro Antimicrobial Activity of Ibuprofen, Naproxen, Aspirin, Amikacin, Aztreonam, and Ceftazidime against P. aeruginosa and E. meningoseptica

*P. aeruginosa* is the most common cause of infection in CF lungs. *E. meningoseptica* has been identified as a newly emerged multidrug-resistant pathogen [34,35]. Hence, we decided to proceed with PA HP3 and EM 2-18 to explore the potential of combination therapy of NSAIDs and antimicrobials. We determined the minimum inhibitory concentration (MIC) of ibuprofen, naproxen, and aspirin against PA HP3 and EM 2-18 (Table 4), amikacin against EM 2-18, and aztreonam and ceftazidime against PA HP3 (Table 5) using the broth microdilution method. The MIC of ibuprofen against PA HP3 is 512 μg/mL, and against EM 2-18, 256 μg/mL, which is consistent with our previous observation that ibuprofen has antimicrobial activity against *P. aeruginosa* laboratory strains and clinical isolates [18]. The MIC of naproxen is 1024 μg/mL against EM 2-18. However, we could not detect the MIC of naproxen against PA HP3 and the MIC of aspirin against PA HP3 and EM 2-18

within our concentration upper limit, which was 1024 µg/mL. Other studies have observed that the MICs of naproxen and aspirin are above 1024 µg/mL against Gram-negative bacteria [36,37]. The MICs of aztreonam and ceftazidime against PA HP3 are 4 and 16 µg/mL, respectively. The MIC of amikacin against EM 2-18 is 16 µg/mL.

**Table 4.** The median minimum inhibitory concentration of ibuprofen, naproxen, and aspirin against PA HP3 and PA 2-18. ND: above > 1024 µg/mL.

| MIC µg/mL Strain | Ibuprofen | Naproxen | Aspirin |
|---|---|---|---|
| PA HP3 | 512 | ND | ND |
| EM 2-18 | 256 | 1024 | ND |

**Table 5.** Fractional inhibitory concentrations of combining aztreonam, ceftazidime, and amikacin with ibuprofen, naproxen, and aspirin against selected isolates of *P. aeruginosa* (PA HP3) and *E. meningoseptica* (EM 2-18). The median MIC values are shown.

| Drug Combination | MIC of Single Drug (µg/mL) | MIC in Combination (µg/mL) | FICI |
|---|---|---|---|
| Aztreonam and Ibuprofen (PA HP3) | 4/512 | 2/50 | 0.60 |
| Ceftazidime and Ibuprofen (PA HP3) | 16/512 | 4/50 | 0.35 |
| Amikacin and Ibuprofen (EM 2-18) | 16/256 | 12/50 | 0.95 |
| Aztreonam and Naproxen (PA HP3) | 4/>1024 | 2/50 | 0.55 |
| Ceftazidime and Naproxen (PA HP3) | 16/>1024 | 4/50 | 0.30 |
| Amikacin and Naproxen (EM 2-18) | 16/>1024 | 12/50 | 0.80 |
| Aztreonam and Aspirin (PA HP3) | 4/>1024 | 4/100 | 1.10 |
| Ceftazidime and Aspirin (PA HP3) | 16/>1024 | 16/100 | 1.10 |
| Amikacin and Aspirin (EM 2-18) | 16/1024 | 16/100 | 1.10 |

*3.3. Synergistic Effects of Ibuprofen and Ceftazidime against a Multi-Drug Resistant Clinical Isolate of P. aeruginosa (PA HP3) Demonstrated by Checkerboard Assay*

To explore the potential synergistic antimicrobial effects between antibiotics and NSAIDs, we tested combined drugs against PA HP3 and EM 2-18 using a checkerboard assay. The MICs of the antibiotics were reduced in the presence of 50 µg/mL ibuprofen and various concentrations of naproxen (Table 3). However, the MIC of the antibiotics did not change with the presence of the highest tested concentration of aspirin, which is 100 µg/mL. We used the fractional inhibitory concentration (FIC) to interpret potential drug combination effects (Table 1). Based on the FIC calculation [22], we determined that ceftazidime is synergistic with ibuprofen against PA HP3. The antimicrobial activity of ibuprofen is additive with aztreonam against PA HP3 and with amikacin against EM 2-18 (Table 4). Similarly, the antimicrobial activity of naproxen is additive with all three antibiotics (Table 4). The combinational MICs are determined based on the turbidity of the liquid assay in 96-well plates. Turbidity measurements have poor sensitivity, which is a limitation

of the checkerboard assay. Hence, we decided to perform a 24 h endpoint colony forming unit (CFU) study to further examine the potential for synergistic drug combinations.

### 3.4. Synergistic Effects of NSAIDs and Antibiotics Demonstrated by Endpoint CFU Studies

Given that we observed synergy between ceftazidime and ibuprofen against PA HP3 using a checkerboard assay, we further explored the possibility of synergy between ibuprofen and standard-of-care antibiotics using an endpoint CFU study. The concentrations used in the 24 h endpoint CFU study were selected based on the checkerboard assay result. For the 24 h endpoint CFU study, we selected NSAIDs and antibiotics concentrations at sub or at individual MIC concentrations but included the combinational MIC within the testing range. The bacterial concentration of PA HP3 is ~$10^9$ CFU/mL when treated with 2 µg/mL aztreonam, or half the MIC of 4 µg/mL. However, following supplementing aztreonam with 100 µg/mL naproxen, the bacterial burden of PA HP3 is reduced to ~$10^6$ CFU/mL (Figure 1A). Since the synergistic effect in an endpoint CFU study is defined as a $\geq$2-$\log_{10}$ reduction in bacterial burden compared with the most efficacious individual treatment, this combination of 2 µg/mL aztreonam and 100 µg/mL naproxen demonstrated synergy. When we treated PA HP3 with 8 µg/mL ceftazidime alone, that is, half the MIC of 16 µg/mL, the bacterial concentration of PA HP3 was ~$10^7$ CFU/mL. When we added 100 µg/mL naproxen, the bacterial burden was reduced to ~$10^4$ CFU/mL, which indicated synergy (Figure 1B). Furthermore, we verified that ibuprofen was synergistic with aztreonam and ceftazidime against PA HP3, and with amikacin against EM 2-18. With the addition of 100 µg/mL ibuprofen, 1 µg/mL aztreonam achieved ~6-$\log_{10}$ CFU/mL reduction compared to individual treatment (Figure 2A); 2 µg/mL ceftazidime achieved ~4-$\log_{10}$ bacterial burden reduction compared to individual treatment (Figure 2B); and 4 µg/mL amikacin achieved ~3-$\log_{10}$ reduction compared to individual treatment (Figure 2C). Thus, both naproxen and ibuprofen demonstrated synergy in combination with aztreonam, ceftazidime, and amikacin. Furthermore, ibuprofen demonstrated a greater reduction in the bacterial burden when combined with all three antibiotics compared with naproxen. Because high-dose ibuprofen has been used in CF patients as an anti-inflammatory drug, we decided to test the efficacy of antibiotics combined with ibuprofen in a murine pneumonia model.

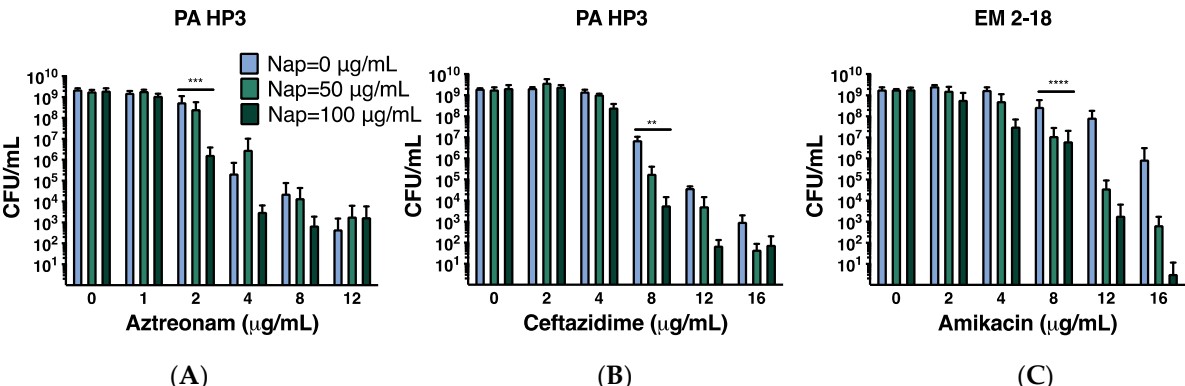

(**A**)           (**B**)           (**C**)

**Figure 1.** Endpoint CFU study demonstrates synergy between naproxen and antibiotics. Synergy demonstrated between naproxen and (**A**) aztreonam, (**B**) ceftazidime, and (**C**) amikacin against PA HP3 and EM 2-18 by endpoint CFU study after 24 h incubation with the drug concentration ratios (in µg/mL) indicated under each panel. Data are shown as mean and standard deviation (n = 6). Statistical significance determined by one-way ANOVA followed by Tukey's multiple comparison test (** indicates $p \leq 0.01$, *** indicates $p \leq 0.001$, and **** indicates $p \leq 0.0001$).

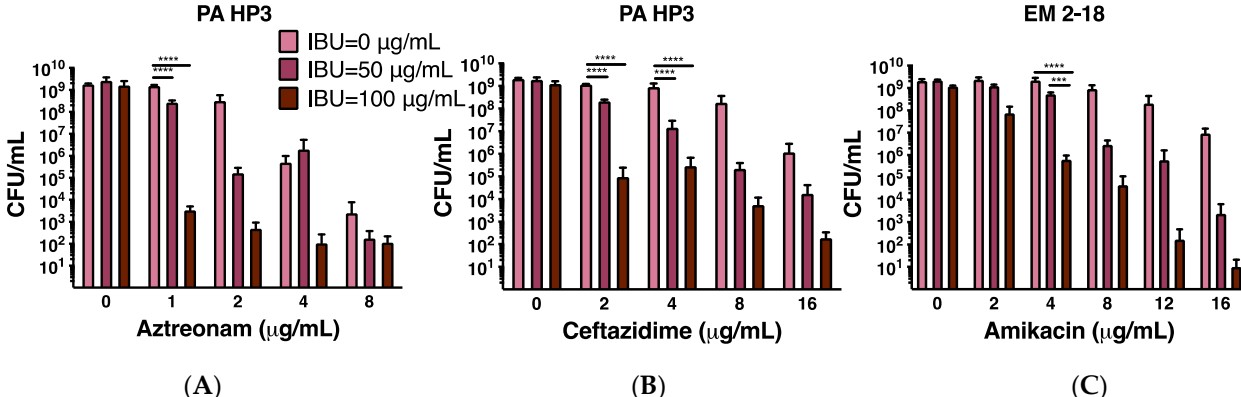

(**A**)                                    (**B**)                                    (**C**)

**Figure 2.** Endpoint CFU study demonstrates synergy between ibuprofen and antibiotics. Synergy demonstrated between ibuprofen and (**A**) aztreonam, (**B**) ceftazidime, and (**C**) amikacin against PA HP3 and EM 2-18 by endpoint CFU study after 24 h incubation with the drug concentration ratios (in µg/mL) indicated under each panel. Data are shown as mean and standard deviation (n = 6). Statistical significance determined by one-way ANOVA followed by Tukey's multiple comparison test (*** indicates $p \leq 0.001$, and **** indicates $p \leq 0.0001$).

### 3.5. Ibuprofen in Combination with Ceftazidime Significantly Improved Mice Survival in an Acute Pneumonia Model

We previously established a pseudomonal in vivo pneumonia model with demonstrated success [38,39]. Based on CLSI standards, PA HP3 is defined as resistant to ceftazidime. Hence, we decided to use the combination of ceftazidime and ibuprofen against PA HP3 to explore the efficacy of combination therapy in the acute murine pneumonia model. To investigate the significance of combinational therapy in an acute pneumonia infection model, mice were intranasally infected with PA HP3 and treated with control, ceftazidime via intraperitoneal injection only, ibuprofen via oral feeding only, and ceftazidime via intraperitoneal injection combined with ibuprofen via oral feeding. Mice were monitored for weight loss, clinical scores, and survival. Infected mice were treated every 8 h for up to 7 doses over 72 h. Infected mice treated with combination therapy demonstrated a significant survival advantage over individually treated groups of mice (Figure 3).

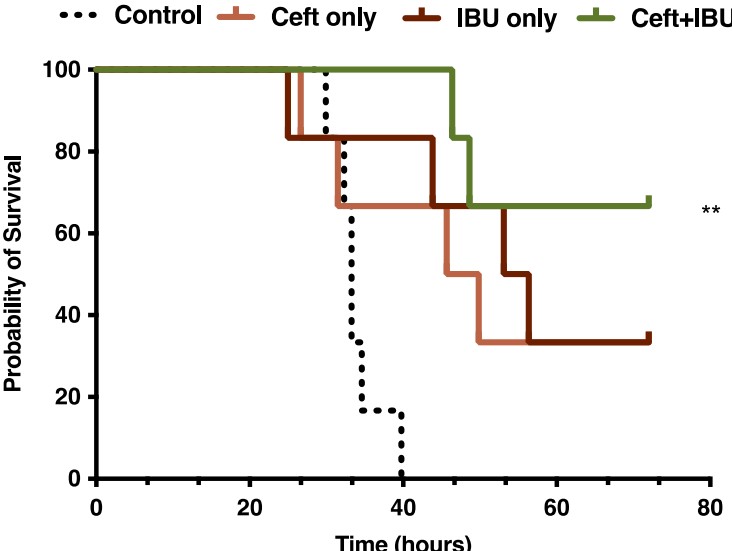

**Figure 3.** The addition of high-dose ibuprofen to ceftazidime improves survival in a pneumonia model. The combinational therapy of ibuprofen and ceftazidime resulted in a significant survival advantage for mice intranasally inoculated with the *P. aeruginosa* isolate PA HP3 (n = 6). Statistical significance was determined by Mantel–Cox test (** indicates $p \leq 0.01$).

## 4. Discussion

The use of high-dose ibuprofen for the treatment of cystic fibrosis (CF) lung disease has well-documented beneficial effects, which have been attributed to its anti-inflammatory activity [39]. We previously found that high concentrations of ibuprofen, equivalent to the peak serum concentrations achieved with high-dose ibuprofen, exhibit antimicrobial effects against pathogens commonly found in CF lungs [39]. Studies have documented that NSAIDs, including ibuprofen, naproxen, and aspirin, synergize with antimicrobials against Gram-positive bacteria [19,40]. However, only a few studies have investigated whether NSAIDs are synergistic with commonly used antimicrobials against pathogens that typically infect the CF lungs. Here, we determined the antimicrobial activity of ibuprofen and other NSAIDs in combination with antibiotics used against pathogens in CF lungs. We show that ibuprofen synergizes with antibiotics in vitro and enhances antimicrobial efficacy in vivo.

We initially screened one clinical isolate of each of five bacterial species commonly found in CF lungs, all considered multi-drug resistant based on CLSI cut points [32]. After the addition of ibuprofen to the agar, several antibiotics demonstrated significant increases in the zones of inhibition compared to a DMSO control. The antibiotics that exhibited increases in the zones of inhibition with ibuprofen were specific to each isolate; that is, no single combination demonstrated enhanced efficacy against all the isolates. We implemented the disc diffusion assay as a rapid screening method to first determine whether ibuprofen could enhance the antimicrobial activity of commonly used antibiotics, and if so, to then identify the antibiotics that show enhanced activity in combination with ibuprofen. We found that ibuprofen enhanced the zones of inhibition around aztreonam, ceftazidime, and amikacin against one or more of the tested Gram-negative bacterial isolates and around gentamicin and vancomycin against our tested MRSA isolate.

*P. aeruginosa* infections in the lungs of CF patients have been problematic to treat, in part, due to the rise in prevalence of multi-drug resistant pathogens [41]. Studies have reported *E. meningoseptica* as one of the emerging multi-drug-resistant pathogens in CF patients [34,35,42]. The isolates of *P. aeruginosa* and *E. meningoseptica* tested, PA HP3 and EM 2-18, respectively, are both clinical isolates classified as multi-drug resistant based on their antibiotic resistance profiles, and are thus, representative of the problematic isolates complicating the care of CF patients. Hence, we selected the antibiotics identified as having enhanced activity with ibuprofen against PA HP3 and EM 2-18 and used broth microdilution methods to determine the MICs of the antibiotics, ibuprofen, and other NSAIDs including naproxen and aspirin. The MICs suggest that high-dose ibuprofen has antimicrobial activity against CF clinical isolates, as we have previously reported [39]. We could not determine the MICs of naproxen and aspirin within the range of the concentrations tested.

Next, we used the checkerboard assay to determine the antimicrobial activity of the selected antibiotics combined with NSAIDs. We observed that adding either ibuprofen or naproxen, but not aspirin, reduced the MICs of the selected antibiotics. The fractional inhibitory concentration (FIC) was used to determine the effects of combining antibiotics with NSAIDs. In previous MIC experiments, we did not obtain the MICs of naproxen and aspirin; hence, we used the highest concentration tested, 1024 µg/mL, as the MICs for those two drugs to calculate the FIC. Based on the FIC interpretation, we determined that ceftazidime is synergistic with ibuprofen against the *P. aeruginosa* isolate, PA HP3. Aztreonam and amikacin showed additive effects when combined with ibuprofen against PA HP3 and the *E. meningoseptica* isolate EM 2-18, respectively. All three antibiotics showed additive effects after combining with naproxen against PA HP3 or EM 2-18. These results suggest that ibuprofen, at plasma concentrations achieved in CF patients [10], can be synergistic with ceftazidime. Because the checkerboard assay provides data only on the inhibition of bacterial growth, we performed a 24 h endpoint CFU assay to validate our synergy results.

The 24 h endpoint CFU assay demonstrated significant reductions in CFU counts when antibiotics were combined with various concentrations of NSAIDs compared with

the antibiotics alone. The assay demonstrated that naproxen is synergistic with both aztreonam and ceftazidime against PA HP3. Furthermore, the 24 h endpoint CFU assay validated that ibuprofen is synergistic with aztreonam and ceftazidime against PA HP3, and with amikacin against EM2-18. The assay suggested that synergy can be achieved at lower concentrations for both antibiotics and ibuprofen than was shown in the checkerboard assay. In addition, the combinational treatments resulted in a more than 3-$\log_{10}$ CFU/mL reduction, which implied bactericidal activity [23].

Both the checkerboard and 24 h endpoint CFU assays showed synergistic effects. However, our endpoint CFU results showed more robust synergistic effects compared to the checkerboard assay. Checkerboard assays and endpoint CFU assays are both commonly used to evaluate the interactions between different antibiotics when used in combination and provide valuable insights into the potential synergistic effects of antibiotic combinations. However, differences in outcomes between these two assays can arise due to the specific nature of each assay and the conditions under which they are conducted. The goal of the checkerboard is to determine the minimum inhibitory concentration (MIC) of each drug when used alone and in combination [43]. In an endpoint CFU assay, bacterial cultures are exposed to individual drugs and drug combinations over a specific time period. This assay provides information about the rate and extent of bacterial killing by the drugs [44]. In addition, in a checkerboard assay, the incubation conditions are typically static, meaning that the test organisms and antimicrobial agents are undisturbed during incubation. However, 24 h endpoint CFU studies are conducted under dynamic conditions, where the culture is typically incubated with continuous shaking or agitation. These dynamic conditions better mimic the conditions in vivo and improve drug diffusion in the solution. Studies have compared the checkboard and endpoint CFU assay results to identify in vitro synergistic antibiotic combinations [45–49]. Bremmer et al. suggested that checkerboard and endpoint studies show excellent agreement [46]. However, some studies show discrepancies between the two assays [45,47–49]. Because ceftazidime and ibuprofen showed synergy in both checkerboard and endpoint assays, we chose this combination for further testing.

Finally, a mouse model of acute *P. aeruginosa* pneumonia treated with a combination of ceftazidime and high-dose ibuprofen validated our in vitro synergistic results. To treat acute pulmonary exacerbations in CF patients harboring *P. aeruginosa*, the Cystic Fibrosis Foundation recommends intravenous (IV) dosing of ceftazidime of 300 mg/kg/day divided every 8 h, while "typical" doses of 200 to 400 mg/kg/day divided every 6 to 8 h with a maximum of 8 to 12 g/day are reported in the literature [50]. The FDA-approved IV dose of ceftazidime for lung infections caused by *P. aeruginosa* in CF patients, however, is 90 to 150 mg/kg/day divided every 8 h [50]. For a mouse that weighs ~20 g, the equivalent "typical" CF dosing ranges of ceftazidime are 1.33 to 2.66 mg per dose every 8 h, and the FDA-approved equivalent CF doses are 0.6 to 1 mg per dose every 8 h. We administrated 0.5 mg per dose every 8 h, which is less than the lowest dose recommended for anti-pseudomonal therapy in CF. In the Konstan et al. study [10], CF patients received 20 to 30 mg/kg/dose of ibuprofen twice daily, which for a ~20 g mouse is 0.4 to 0.6 mg per dose. We used 0.5 mg ibuprofen per dose, which is in the range of high-dose ibuprofen used for CF patients, and delivered it every 8 h instead of the twice-daily dosing that the CF patients received. Although the MICs for ceftazidime (16 µg/mL) and for ibuprofen (512 µg/mL) against PA HP3 are widely divergent, the low ceftazidime dose (75 mg/kg/day IP divided q8; 17% of the highest "typical" CF dose) and the high ibuprofen dose (75 mg/kg/day orally divided q8; 125% of the highest CF "high-dose") delivered to the two single-drug groups resulted in a similar survival rate of ~33%. Previously, although the results were not statistically significant, Konstan et al. showed that CF patients in the ibuprofen group had fewer hospital admissions and shorter hospital stays than CF patients in the placebo group [10]. In addition, high-dose ibuprofen reduced inflammation in the airways after acute endotoxin challenge [14] and long-term challenges [51]. We have shown that after a lethal inoculum of the MDR-*P. aeruginosa* isolate, PA HP3, the combination of

ceftazidime and ibuprofen significantly enhanced survival compared with other groups. Thus, the combination of high-dose ibuprofen with a standard-of-care anti-pseudomonal antibiotic appears to synergistically increase mouse survival in an acute pneumonia model. Taken together, NSAIDs alone and in combination with standard-of-care antibiotics may improve treatment outcomes for both acute pulmonary exacerbations and chronic infections in CF patients.

One limitation of this study is that the disc diffusion assay has been reported to be relatively insensitive and, in some cases, inaccurate [52]. In particular, the disc diffusion assay for vancomycin is prone to significant errors, prompting the CLSI recommendation that MIC testing follow disc diffusion screens to differentiate vancomycin-susceptible from vancomycin-intermediate isolates [31,53]. Similarly, the disc diffusion assay for colistin is error-prone due to poor diffusion into agar, and currently, no standard disc diffusion assay for colistin exists [54]. Thus, synergistic combinations may have been missed, and some of the enhanced activity seen in our screen may not have been replicated in other assays. The first limitation remains. However, we addressed the second limitation using the checkerboard assay and 24 h CFU study to validate our results. To demonstrate proof of principle, we only tested combinations against isolates from *P. aeruginosa* and *E. meningoseptica*. We did not test all the promising combinations from the disc diffusion assay against other bacterial strains. Although we only used one isolate from each of the two bacterial species to validate our initial findings, our results suggest that this approach may be applicable to other isolates and bacterial species found in the lungs of CF patients. We acknowledge that CF patients suffer from chronic lung infections with bacteria growing in biofilms. We have previously found that the concentrations of ibuprofen used clinically in CF reduce *P. aeruginosa* biofilm accumulation [39]. It would be interesting to test whether ibuprofen changes the activity of antibiotics against bacteria growing in biofilms, both in vitro and in vivo. We would again expect to see enhanced antimicrobial activity of antibiotics in combination with ibuprofen.

To summarize, ibuprofen significantly increases the zone of inhibition of standard-of-care antibiotics against CF clinical isolates in a disc diffusion assay. Combinations exhibiting increased zones of inhibition compared with the antibiotics alone identified in the disc diffusion assay represented candidates for validation in further microbiological assays. In vitro studies suggested that ibuprofen has antimicrobial activity against CF clinical isolates, which confirmed our previous observation [38]. Although aspirin did not demonstrate either synergy or additive effects in combination with the selected antibiotics, aztreonam, ceftazidime, and amikacin, both naproxen and ibuprofen showed additive effects, and ibuprofen showed synergistic effects with one or more of the antibiotics. Further, the 24 h endpoint CFU study confirmed that naproxen and ibuprofen are synergistic with at least one of the three tested antibiotics. Ibuprofen in combination with aztreonam or ceftazidime exerted a greater reduction in the bacterial burden than did naproxen. Finally, mice intranasally inoculated with *P. aeruginosa* isolate PA HP3, and treated with sub-MIC concentrations of ceftazidime intraperitoneally and ibuprofen orally, demonstrated improved survival rates compared with single-drug treated mice. Our in vitro and in vivo experiments suggest that high-dose ibuprofen has antimicrobial activity in addition to its anti-inflammatory properties, particularly when combined with standard-of-care antibiotics. Currently, high-dose ibuprofen treatment is not widely available in CF centers [6,7]. We suggest that it would be worth revisiting high-dose ibuprofen to explore its clinical benefits in CF patients.

**Author Contributions:** Conceptualization, Q.C., K.N.S. and C.L.C.; methodology, Q.C. and C.L.C.; data collection, Q.C., M.I., S.B.S. and B.C.; data analysis, Q.C. and C.L.C.; Q.C., M.I., S.B.S., B.C., K.N.S. and C.L.C. wrote and revised the manuscript. All authors have read and agreed to the published version of the manuscript.

**Funding:** This project was supported by the Men of Distinction foundation.

**Institutional Review Board Statement:** Not applicable.

**Informed Consent Statement:** Not applicable.

**Data Availability Statement:** Not applicable.

**Acknowledgments:** This study appears in the doctoral dissertation of Qingquan Chen, which was conducted at the Texas A&M University Health Science Center and supervised by Carolyn Cannon.

**Conflicts of Interest:** The authors declare no conflict of interest.

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
