# Peer review of "Synergistic Antimicrobial Effects of Ibuprofen Combined with Standard-of-Care Antibiotics against Cystic Fibrosis Pathogens"

_biomedicines, doi:10.3390/biomedicines11112936_

Round 1
Reviewer 1 Report (Previous Reviewer 2)
Comments and Suggestions for Authors
The manuscript describes the antimicrobial activity of ibuprofen and other NSAIDs in combination with antibiotics used against CF lungs pathogens.
There are a few small corrections I believe should be made:
1) Line 89, The name of the bacterium used for the first time in the manuscript should be written without abbreviations (Achromobacter xylosoxidans).
2) Lines 230 – 232, The authors claim that "Particularly, three strains, PA HP3, EM 2-18, and HI 2501 indicated in orange were non-susceptible to three classes of antibiotics, and thus, met the definition of multi-drug resistant" while according to data in Table 2, P. aeruginosa showed resistance to two classes - beta-lactams (ceftazidime and aztreonam) and aminoglycosides (tobramycin).
3) Line 245, instead of EA 2-18 there should be EM 2-18.
4) Lines 245 – 247, the finding that "Gentamicin and vancomycin, with the addition of 100 μg/ml ibuprofen, showed significant increases in the zones of inhibition against SA LL06 (Table 3)." is too far-fetched. An increase in the zone of 1 - 2 mm is not a significant increase.
5) Tabela 5, According to the data in Table 4, aspirin MIC for EM 2-18 (last line) should be > 1024
6) Line 292, instead of Table 4 there should be Table 1.
Author Response
Reviewer 1
The manuscript describes the antimicrobial activity of ibuprofen and other NSAIDs in combination with antibiotics used against CF lungs pathogens.
We thank the reviewer for their supportive comments and time and effort in reviewing this manuscript. We have revised the manuscript to address the comments and suggestions, and we hope that it is now acceptable for publication.
There are a few small corrections I believe should be made:
- Line 89, The name of the bacterium used for the first time in the manuscript should be written without abbreviations (Achromobacter xylosoxidans).
We thank the reviewer for this comment. We have updated the name of the bacterium in the text.
- Lines 230 – 232, The authors claim that "Particularly, three strains, PA HP3, EM 2-18, and HI 2501 indicated in orange were non-susceptible to three classes of antibiotics, and thus, met the definition of multi-drug resistant" while according to data in Table 2, P. aeruginosa showed resistance to two classes - beta-lactams (ceftazidime and aztreonam) and aminoglycosides (tobramycin).
We thank the reviewer for this comment. The definition of MDR bacteria that we followed to categorize the isolates is that of the Centers for Disease Control (CDC): https://www.cdc.gov/nhsn/pdfs/ps-analysis-resources/phenotype_definitions.pdf. However, the CDC does not mention monobactams, such as aztreonam, in the definition of MDR--P. aeruginosa. The non-susceptibility of PAHP3 to aztreonam could be lumped together with the resistance to cephalosporins, as you suggest, but experts debate that interpretation. We have chosen in this case to rely on the work of Magiorakos et al. (see Table 4) [30, doi: 10.1111/j.1469-0691.2011.03570.x].
These authors, who include representatives from the CDC, list monobactams in a separate category for determining MDR for P. aeruginosa rendering PAH3 non-susceptible to 3 classes and MDR. The text has been updated to clarify this point.
- Line 245, instead of EA 2-18 there should be EM 2-18.
We thank the reviewer for this comment. We have updated the correct strain name.
- Lines 245 – 247, the finding that "Gentamicin and vancomycin, with the addition of 100 μg/ml ibuprofen, showed significant increases in the zones of inhibition against SA LL06 (Table 3)." is too far-fetched. An increase in the zone of 1 - 2 mm is not a significant increase.
We thank the reviewer for this comment. The experiments were repeated in triplicate, and the median zone of inhibition was reported. When comparing the average zone of inhibition diameter for gentamicin and vancomycin with the addition of 100 μg/ml ibuprofen, we observed statistical significance between the two conditions. We agree that there is only 1-2 mm difference in the reported median zone of inhibition and have updated the text.
- Table 5, According to the data in Table 4, aspirin MIC forEM 2-18 (last line) should be > 1024
We thank the reviewer for this comment. We have updated the aspirin MIC for EM 2-18 to > 1024.
- Line 292, instead of Table 4 there should be Table 1.
We thank the reviewer for this comment. We have updated as Table 1.

Reviewer 2 Report (Previous Reviewer 1)
Comments and Suggestions for Authors
This manuscript focuses on the combination of NSAIDs with antibiotics against CF pathogens, in vitro and in a mouse infection model.
This is a resubmission of a previoulsy submitted manuscript which I reviewed for the journal. Most of my previous concerns were adressed in the present resubmission, and the discussion part has been greatly improved.
There remains two points for which I would appreciate further explanations:
- Table 1/line 155. In the reference cited by the authors, there is no difference made between “additive” and “indifferent” interaction zones (they even call the 0.5-4 FICI zone the “additive/indifferent” zone). How was the difference made in the present study, and what is its significance?
- Table 3 versus Table 2. For strain PA HP3: There is a major decrease in Amikacin diameter when adding DMSO (19mm without DMSO, 8mm with DMSO). How do the authors explain this result? If DMSO and amikacin were antagonistic, such a decrease in amikacin diameter should be observed for the other strains submitted to amikacin+DMSO; on the other hand, if DMSO alone explains the difference, it should be observed with other drugs against PA HP3.
Minor comments:
- line 258. In Table 3, the three diameters with an * indicate a =5mm increase instead of a >5mm increase. Therefore the legend should be: “* indicates a ≥5mm increase”.
Author Response
Reviewer 2
This manuscript focuses on the combination of NSAIDs with antibiotics against CF pathogens, in vitro and in a mouse infection model.
This is a resubmission of a previously submitted manuscript which I reviewed for the journal. Most of my previous concerns were addressed in the present resubmission, and the discussion part has been greatly improved.
We thank the reviewer for the supportive comments and time and effort in reviewing this manuscript. We have revised the manuscript to address the comments and suggestions, and we hope that it is now acceptable for publication.
There remains two points for which I would appreciate further explanations:
- Table 1/line 155. In the reference cited by the authors, there is no difference made between “additive” and “indifferent” interaction zones (they even call the 0.5-4 FICI zone the “additive/indifferent” zone). How was the difference made in the present study, and what is its significance?
We thank the reviewer for this comment. The definition of indifference, in which the combination of antibiotics is equal to the effects of the most active antibiotic, is FIC index above 1. In order to distinguish the difference between additive and indifferent effects, we use 0.5 < FIC ≤ 1 to define the additive effect and 1 < FIC ≤ 4 to define the indifferent effect. This FIC index has also been used by other studies, which are updated in the citations. We acknowledge that if one drug is more active than another, the indifference effect is hard to differentiate from an additive effect. However, we didn’t observe such in our results.
- Table 3 versus Table 2. For strain PA HP3: There is a major decrease in Amikacin diameter when adding DMSO (19mm without DMSO, 8mm with DMSO). How do the authors explain this result? If DMSO and amikacin were antagonistic, such a decrease in amikacin diameter should be observed for the other strains submitted to amikacin+DMSO; on the other hand, if DMSO alone explains the difference, it should be observed with other drugs against PA HP3.
We thank the reviewer for this comment. We double-checked our original results and realized it was caused by a mistake while tabulating the results into the table. We apologize for the confusion and mistake. We have updated the median zone of inhibition diameter in table 3. For AMK, without ibuprofen, the median zone of inhibition diameter is 19 mm. With ibuprofen, the median zone of inhibition diameter is 21 mm.
Minor comments:
- line 258. In Table 3, the three diameters with an * indicate a =5mm increase instead of a >5mm increase. Therefore, the legend should be: “* indicates a ≥5mm increase”.
We thank the reviewer for this comment. We have updated the legend in Table 3.

Round 2
Reviewer 2 Report (Previous Reviewer 1)
Comments and Suggestions for Authors
All previous points were adressed by the authors in the new version of the manuscript.
This manuscript is a resubmission of an earlier submission. The following is a list of the peer review reports and author responses from that submission.
Round 1
Reviewer 1 Report
Comments and Suggestions for Authors
This manuscript focuses on the combination of NSAIDs with antibiotics against CF pathogens, in vitro and in a mouse infection model.
The manuscript is well written, but maybe some results could be presented more clearly. Moreover, some further details and explanations are needed for several experiments. The discussion part should be improved as it mostly repeats what was shown in the results part:
- Line 111 and 121. Collection/Lab control strains (e.g. ATCC strains or PAO1 lab strain) should have been added for MIC and disc diffusion assays.
- Table 1. Please add a reference to justify the difference between “additive” and “indifferent” zones
- Table 2. For colistin and vancomycin: Variations in susceptibility testing are difficult to observe using disc diffusion due to the difficulty for both these drugs to diffuse correctly in the solid medium. In general, disc diffusion is rather difficult to interpret for clinical strains isolated from CF patients, especially after a simple 24-hour incubation.
- Figure 1. (i) Diameters are difficult to read using a graph (e.g. ATM for PA HP3: is it 16, 17, or 18mm?). A more common way (and more understandable to clinical microbiologists) to compare diameter would be to show the median diameters obtained from all assays with and without IBU and to see if there is a >5mm difference, since a >5mm difference is commonly considered as significant using disc diffusion. (ii) Why are results shown only for some drugs? Was not the same drug panel (ATM, CAZ, and AMK) tested for all GNB? (iii) The IBU concentration used here should appear in figure legend.
- Line 136 and 140,141,142. There is a 0µg/mL well for NSAIDs but not for ATBs? There should be a 0:0 µg/mL well in each plate to serve as positive control.
- Lines 160-163 and Fig 2 -3. There again, no 0 µg/mL control well for each ATB?
- Table3, Table4. According to the method part, these experiments were performed in triplicate and duplicate, respectively. What are the numbers shown (Mean? Median?)? Please be more specific in Tables legends. This is of importance, expecially for results showing significant synergy.
- Lines 270-272. FICindex results should be shown. There again, a more reader-friendly presentation of checkerboard assays would be to show in a same table MICs of single drugs, MICs of combinations, and FICI. See e.g. the presentation of Table 2 in Antimicrob Agents Chemother. 2019 Apr; 63(4): e02613-18.
- Line 373-374. The authors often refer to “high dose ibuprofen”. This is rather vague for a non-ibuprofen specialist, to which plasma concentration would it correspond in a CF patient? What daily oral dose?
- It would have been interesting to study synergy using a more “traditional” time-kill assay protocol: starting inoculum of ca. 10^7 UFC/mL, and showing more counting times (e.g 4h, 8h, 12h, 24h, 48h). Maybe in a next study?
- Fig 4. (i) More details are needed here: Were there six mouses in each experiment? How many times was the experiment repeated? What does the 60% survival stand for (60%*6=3.6 mice at 72h for the Ceft/IBU group?)? (ii) Moreover, I do not understand the graph scales: were the mouses monitored 24/7, since in the control group mice seem to die every 30 minutes? And some survival percentages seem very precise for groups of only six mice (at H46, the IBU group has ca. 83% survival versus 80% survival for the ceft+IBU group…).
- Discussion part: Please further discuss the three following points: (i) Time-kill results seem to show much more synergistic combinations than what was shown by checkerboard assays, do the authors have an explanation? (ii) Figure 4: the percentage of survival is the same using ceftazidime only and IBU only, please discuss, with a focus on the wide difference in MICs of both drugs; (iii) Figure 4: IBU only seems to reduce mortality at 72h in comparison to the control group; I am well aware of studies discussing NSAIDs effects upon the chronic condition of CF patients, but are there also data on acute pulmonary exacerbations/acute infection episodes?
Minor comments:
- Line 96. Bacteria names should be written in italics.
- Line 118. Please give each antibiotic disc content (in µg).
- Line 11—120. Which guidelines were used? CLSI?
- Line 220-227.”high dose ibuprofen” or “100µg/mL ibuprofen” are repeated several times: Was it not the same ibuprofen concentration for all disc diffusion experiments? What does “high dose” stand for?
- Line 152. “Determination of bacterial burden for synergistic drug combinations”: I think “Time-kill assays” would be a more reader-friendly title.
- Line 250-251. Please give a reference, or move this statement in the discussion part.
- Line 252. MIC of naproxen is 1024 in Table 3?
Reviewer 2 Report
Comments and Suggestions for Authors
The paper entitled Synergistic Antimicrobial Effects of Ibuprofen Combined with 2 Standard-of-Care Antibiotics against Cystic Fibrosis Pathogens fits the scope of the journal Biomedicines.
The topic of manuscript is an interesting contribution to scientific knowledge in the field of antimicrobials.
However, I have some concerns about the manuscript submitted.
1) Section 2.6. The determination of synergistic drug combinations is not described in sufficient detail.
2) Table 3 and rows 252 and 254. Inconsistency of data in the table and in the description of the results.
3) No results for DMSO in diffusion method.
4) Section 2.2. Bacterial strains. The names of microorganisms should be written in italics.